# Considerations for Satisfactory Sedation during Dental Implant Surgery

**DOI:** 10.3390/jpm13030461

**Published:** 2023-03-01

**Authors:** Takaya Ito, Nozomi Utsumi, Yukiko Baba, Tomoka Matsumura, Ryo Wakita, Shigeru Maeda

**Affiliations:** Department of Dental Anesthesiology and Orofacial Pain Management, Tokyo Medical and Dental University, 1-5-45 Yushima, Bunkyo-ku, Tokyo 113-8510, Japan

**Keywords:** literature review, conscious sedation, nerve block, local anesthetic, vasoconstrictor agents

## Abstract

Implant surgery is a lengthy dental procedure, and sedation is often used to reduce discomfort. The effectiveness of sedation has traditionally been evaluated in terms of patient and surgeon satisfaction, but the most important goal is not to induce a deep sleep in the patient, but rather to ensure that the surgery is performed safely and as planned. Additionally, adequate pain control is a necessary requirement for patient and surgeon satisfaction. Most patients undergoing implant surgery are middle-aged or older, and a relatively large number of them have cardiovascular disease. Infiltration anesthesia using articaine or lidocaine in combination with adrenaline is widely used, but its use in patients with cardiovascular disease is limited because of adrenaline’s effects on the cardiovascular system. The use of long-acting local anesthetics and the potential efficacy of ultrasound-guided jaw nerve block have been investigated to enhance analgesia without resorting to adrenaline. Midazolam and propofol are usually used for sedation, but dexmedetomidine, which causes less respiratory depression, and the ultrashort-acting benzodiazepine remimazolam are emerging as potential alternatives. Monitoring of anesthetic depth using electroencephalography is effective in maintaining a constant level of sedation. In addition, sedation promotes the stabilization of heart rate and blood pressure, reducing the risks associated with adrenaline and allowing for safer management.

## 1. Introduction

Implant surgery involves many different types of procedures in addition to simple placement. When extensive and lengthy surgery is required, sedation is often used to reduce discomfort. Even with sedation, the local anesthetic must be sufficiently effective to ensure patient satisfaction. Studies of patient and surgeon satisfaction with sedation have shown that longer treatment times tend to decrease patient satisfaction. A study of third molar extractions showed that the procedures took an average of 21 min, and both operators and patients reported high satisfaction levels (8–9 on a 0–10 scale) [1]. In third molar extractions with an average sedation time of less than 60 min, over 75% (75.9%) of dentists and 70% (71.2%) of patients rated their experience as “good” on a 4-point scale (good, fair, poor, or very poor) [2]. For implant procedures with an average sedation time of 90 min, 90% of surgeons reported “adequate sedation”, while only 34.4% of patients reported “agreeable” [3]. This may be related to the wearing-off of the effects of local anesthetics for a longer duration of operation.

Several anesthetics are currently used for sedation during implant surgeries. Each has its own characteristics, and combining the advantages of each anesthetic is considered to improve anesthesia management. Patients undergoing implant surgery are often at a relatively high age, and these patients have systemic diseases at a higher rate. Systemic disease, such as cardiovascular diseases, limits the use of vasoconstrictors added to local anesthetics [4,5,6]. The metabolism of some sedatives is affected by other drugs. Obesity affects the effects and body clearance of sedatives. The consideration for these factors of each operation in each patient is expected to lead to better management of pain and discomfort during implant surgery. This review focused on local anesthesia and sedation and the factors related to them.

## 2. Pain Control by Local Anesthesia for Implant Surgery

### 2.1. Consideration of Vasoconstrictors

It is essential to provide patients with pain-free surgery for which local anesthetic agents such as lidocaine, mepivacaine, prilocaine, and articaine are currently widely used. Adrenaline is often added as a vasoconstrictor to lidocaine and articaine cartridges, which enhances the effect of anesthetics, reduces the dose needed, prevents the rapid transfer of the anesthetic into the bloodstream, and clarifies the surgical area. A review reported that 4% articaine with 1:100,000 adrenaline was more effective than 2% lidocaine with 1:100,000 adrenaline [7,8]. Additionally, 2% lidocaine with 1:100,000 adrenaline was superior to 3% prilocaine with 0.03 IU felypressin [9]. A meta-analysis of the effects of local anesthetics on the extraction of mandibular third molars reported that 4% articaine was the most effective [10].

However, because of the adverse effects of local anesthesia, such as tachycardia and arrhythmia, which are caused by the activation of alpha-1 and beta-1,2 adrenaline receptors, prilocaine with felypressin is superior in controlling heart rate [11]. Therefore, prilocaine with felypressin may be better for patients with coronary artery disease, as tachycardia is an associated risk factor in such patients. The inclusion of methylparaben in prilocaine cartridges may induce allergic reactions at a higher rate than those without methylparaben [12], which may be a reason for the limited use of prilocaine with felypressin. Although mepivacaine is inferior in terms of strength, hemodynamic changes after injection are limited [13], indicating that mepivacaine without vasoconstriction is better for patients with cardiovascular diseases [14]. Methemoglobinemia is induced by prilocaine and benzocaine [15]. O-Toluidine, a metabolite of prilocaine, can oxidize ferrous (Fe^2+^) hemoglobin to ferric (Fe^3+^) hemoglobin, which cannot bind and transport oxygen. Therefore, 8 mg/kg prilocaine is generally accepted as the maximum dose to prevent methemoglobinemia [16].

The concentration of adrenaline added to local anesthetics is a significant factor that affects both anesthetic and side effects. In a randomized clinical trial (RCT) comparing the addition of adrenaline to articaine in inferior alveolar nerve block, the extraction of mandibular teeth could be performed without adrenaline, while a longer effect was obtained when using an anesthetic with adrenaline [17]. However, although adrenaline is added to enhance the effect of local anesthetics, it can be toxic to some patients with adrenaline sensitivity due to cardiovascular diseases and/or changes with aging [4,5,6]. In addition, since most patients undergoing implant surgery are older, sensitivity to adrenaline is considered to be higher. However, in a study that compared the dilution of adrenaline added to lidocaine at 1:80,000 with 1:200,000 in healthy adults scheduled for bilateral wisdom tooth extractions, no predominant change in heart rate or mean blood pressure was induced by using lidocaine with adrenaline at 1:200,000, while the anesthetic effects were comparable with lidocaine with adrenaline at 1:80,000 [18]. In addition, there was no statistically significant difference in success or failure in the effect of infiltration injection of local anesthesia at adrenaline concentrations of 1:50,000, 1:80,000, or 1:100,000 [19]. Therefore, the dilution of adrenaline is likely to be effective in controlling its adverse side effects.

### 2.2. Clues for Prolongation of the Effect of Local Anesthesia

The duration of local anesthesia is a crucial factor in dental procedures, and while it may not meet the requirements for caries and pulpal treatment, it is advantageous for implant surgery, which requires long-term pain management. Ropivacaine and bupivacaine are viable options for implant surgery as they have higher lipid solubility than lidocaine and possess inherent vasoconstrictive properties that allow for longer-lasting effects. In an RCT, ropivacaine was found to be more effective than lidocaine with adrenaline in terms of both duration and quality of anesthesia for implant surgery [20]. Additionally, a meta-analysis showed that 0.5% bupivacaine with 1:200,000 adrenaline had a longer duration of effect and a later onset than 2% lidocaine with 1:100,000 adrenaline [21].

In extensive procedures, there is a risk of local anesthetic overdose, especially in patients with compromised liver function [22]. This risk is further increased by the slow metabolism of lidocaine. However, over 90% of articaine is rapidly hydrolyzed by esterases to its inactive metabolite, articainic acid, in the plasma or tissues [23,24,25]. This property results in lower systemic toxicity owing to the rapid systemic excretion of articaine [24], making it safer for patients with hepatic dysfunction and procedures requiring increased doses of local anesthetics. Moreover, articaine has been demonstrated to have the same level of safety as lidocaine in routine dental treatment [26], indicating that it is likely to be more effective and safer than lidocaine.

Liposomal bupivacaine, in which bupivacaine is enclosed in the liposomal membrane, has been developed as a long-acting local anesthetic, and is reported to be effective compared with traditional pain management as a postoperative analgesia of mainly plastic surgery [27]. However, its superiority over the usual infiltrative injection for dental treatment has not yet been established [28,29]. In addition, a systematic review of liposomal bupivacaine for regional operations, mainly used for nerve blocks, could not reach a definitive conclusion regarding whether it is more effective than plain bupivacaine [30]. Thus, liposomal bupivacaine may be more effective for the management of postoperative pain than local anesthesia for implant surgery. 

Conventional nerve block with inferior alveolar nerve block (IANB) using articaine with adrenaline has been reported to be effective against pain caused by drilling for implants in RCT [31], while it has also been reported to be ineffective [32]. The fact that the failure rate of conventional IANB is high [33] may be related to this discrepancy. A meta-analysis showed that intraosseous injection with buccal infiltration anesthesia is more effective than conventional IANB alone for mandibular molar pulpitis [34,35] and that the duration of anesthetic effect is shorter [36]. Thus, intraosseous anesthesia is likely to be effective within the mandible and may be effective for pain caused by drilling but not for prolongation of the anesthetic effect. 

Recently, a novel technique of ultrasound-guided mandibular nerve block (MNB) has been developed, in which a local anesthetic is injected into the lateral pterygoid plate using an extraoral approach [37]. Ultrasound-guided MNB using 5–6 mL of 0.375% ropivacaine enabled the management of general anesthesia for mandibular sequestrectomy with a lower dose of opioid and decreased the need for analgesics for 3 days after surgery [38]. For fixation of mandibular fractures under general anesthesia, ultrasound-guided MNB using 10 mL of 0.5% ropivacaine has been reported to be effective both during and after surgery [39]. Besides, ultrasound-guided MNB has been reported to be safe to perform [40]. Therefore, ultrasound-guided MNB with long-acting local anesthetics is expected to be useful for pain management during and after implant surgery. However, inserting a needle from the inferior aspect of the zygomatic arch to its depth can be intimidating to patients. Therefore, it is crucial to understand their emotional state and provide information about various pain management techniques, allowing patients to make informed choices (Table 1).

## 3. Sedatives

Compared with other dental treatments, implant surgery is invasive and takes a longer time, resulting in stressful situations that people hope to avoid. This is partly why sedation is popular for implant surgery, and another reason is that sedation stabilizes vital signs even if the level is minimal [41]. Since implant surgery requires a higher dose of local anesthetics, including catecholamines, having stable vital signs contributes to safety in implant surgery. Propofol and midazolam are the two main anesthetics used [42]. Dexmedetomidine directly acts on the alpha 2 adrenaline receptor, contributing to the control of changes in vital signs brought about by implant surgery. Remimazolam, the newest ultrashort benzodiazepine sedative, is expected to be useful for sedation in implant surgery. The following paragraphs discuss the characteristics of each sedative and the factors affecting its efficacy.

### 3.1. Propofol

Propofol (2, 6-diisopropylphenol) is highly lipophilic, crosses the blood–brain barrier rapidly [43], and is a short-acting agent with a rapid metabolism, thus enabling rapid recovery from sedation, regardless of sedation depth or length [44]. However, the pharmacokinetic parameters of propofol vary depending on patient factors such as sex [45], obesity [46], cardiac output (CO) [47], and hepatic blood flow [48]. 

With respect to sex, the plasma concentration of propofol decreases more rapidly in females than in males [49], and females tend to recover faster from propofol anesthesia than males [50]. This may because of sex-dependent differences in the formation of liver cytochrome P450s (CYPs), the main metabolic enzymes for propofol [51,52], and UDP-glucuronosyltransferases (UGTs), the main enzymes that catalyze the glucuronidation of propofol [45,53]. To maintain the same level of sedation with propofol between males and females, higher doses are required in females because they metabolize propofol faster than males do [54].

Intravenous sedation in obese patients is relatively difficult to perform compared with that in non-obese patients because airway obstruction easily occurs in obese patients. As the relationship between obesity and sleep apnea is well documented [55], obese patients have a higher risk of respiratory depression during sedation. In addition, the induction time for the same target concentration of propofol is significantly shorter in obese patients [56], suggesting that a lower concentration of propofol is sufficient to sedate and/or that the pharmacokinetics in obese patients differ from those in non-obese patients. 

Age has been shown to affect the efficacy of propofol. When the same dose of propofol is administered during induction of general anesthesia and its effect is examined by EEG changes, it has been shown that the depth of anesthesia is deeper in older patients and that a smaller dose of propofol is sufficient to maintain the depth of anesthesia in elderly patients [57,58].

Although propofol is largely metabolized by the liver [48] and kidneys [59], the size and capacity of both organs in obese patients are equal to those in nonobese patients. This indicates that the rate of propofol metabolism is lower in obese patients, especially when the propofol dose is determined based on the amount per body weight. In addition, obesity may cause fatty degeneration of the liver and/or glomerular injury of the kidneys, possibly leading to a reduction in propofol elimination [60]. Thus, the difficulty in performing sedation increases with increasing body mass index. 

Systemic clearance of propofol decreased by up to 42% in the anhepatic phase in patients revived after reperfusion of the liver during living donor liver transplantation [48]. However, total body clearance was not significantly reduced in patients with liver cirrhosis compared with that in control patients [61,62], suggesting that patients with liver cirrhosis may be able to eliminate propofol via an extrahepatic mechanism. As one-third of the total body clearance of propofol is reportedly shared by the kidneys [63], they are relatively important for the extrahepatic elimination of propofol. In contrast, propofol has been reported to ameliorate liver dysfunction in animal experiments [64], suggesting that propofol may be favorable for sedation in patients with reduced liver function. 

Liver blood flow, but not CO, is a predictive indicator of propofol clearance in critically ill patients [65]. Although liver blood flow changes in response to food intake [66], no relationship was observed between liver blood flow and CO in experiments using normal dogs that ate meals and exercised on a treadmill [67]. This indicates that liver blood flow affects the metabolism of propofol to some extent, independently of CO. Consequently, propofol can be eliminated in patients with deteriorated liver and normal kidney functions. Moreover, if the liver blood flow can be easily measured, it may enable a more accurate prediction of the rate of propofol metabolism. 

As described above, the kidney is another organ responsible for propofol elimination. However, in an experiment involving the induction of general anesthesia in patients with end-stage kidney disease, the effect site concentration of propofol at the time of loss of consciousness was lower, but the difference was not statistically significant [68]. Therefore, a similar or lower dose of propofol is recommended for the induction of general anesthesia in patients with end-stage kidney disease [68]. This is partly because most propofol is metabolized in the liver and the metabolites do not have pharmacological effects. In contrast, another study recommended a higher dose of propofol for the same purpose [69], suggesting that propofol can be safely used in patients with kidney dysfunction. However, increased CO was shown to eliminate plasma propofol in pigs; thus, the lungs and muscles may also contribute to propofol elimination [47]. Therefore, propofol can be safely used in patients with decreased renal function; however, its elimination in patients with reduced liver and kidney function remains unclear. CO is considered to affect the metabolism of rather than renal function.

### 3.2. Midazolam

Although midazolam is a relatively short-acting benzodiazepine, its metabolism and elimination times are longer than those of propofol. The effects of midazolam can be reversed by flumazenil [70], and both respiratory and circulatory depression have been reported to be lower than with propofol [71]. Furthermore, midazolam exhibits a stronger amnesic effect than propofol and dexmedetomidine in healthy individuals [1]. Thus, although midazolam is not a new anesthetic, it still has some advantages and is useful for sedation during dental treatments without requiring a syringe pump.

Obese patients have a higher volume of distribution after midazolam administration than healthy subjects, which suggests the possibility of lower blood midazolam levels after administration and slower recovery [72,73]. The clearance and volume of distribution were similar between elderly and adolescent patients; however, pharmacodynamic data showed significant differences between the two groups, indicating that a lower dose is sufficient to achieve sedation in elderly patients [74,75].

Patients with liver cirrhosis showed a distribution and protein binding comparable to those in healthy controls. However, the elimination time is significantly delayed in patients with cirrhosis; therefore, a lower dose of midazolam is recommended for such patients [76]. In patients with chronic kidney disease, most parameters, such as the free fraction, volume of distribution, and clearance, were higher than those in healthy volunteers [77]. Although the elimination half-life of midazolam was almost identical when the parameters were corrected for protein binding, a lower dose of midazolam was proposed for patients with chronic kidney disease. If additional sedative administration is required, propofol should be recommended. 

Midazolam is metabolized by CYP3A4 into several metabolites, including the active metabolite alpha-hydroxymidazolam [78,79]. Drug interactions can reduce or increase CYP3A4 activity; however, their effects on CYP3A4 vary depending on the medicine, rather than the category of medicines. For example, the area under the curve (AUC) of midazolam is 2.6–8 times that noted after itraconazole, and the AUC after changing itraconazole to rifampicin was only 2.3% of that during itraconazole treatment [80]. In a group of macrolides, pretreatment with clarithromycin increased the AUC of oral midazolam; however, no such effect was observed for azithromycin [81]. Calcium channel blockers are very popular for controlling blood pressure, and both diltiazem and verapamil increase the AUC of oral midazolam 3–4-fold compared with placebos [82]. As described above, midazolam is very sensitive to CYP3A4 and can be used as a probe to determine whether the investigated drug is an inhibitor or inducer of CYP3A4 [79].

According to a study [83], moderate-to-severe disinhibition was observed in 19.5% of patients undergoing bronchoscopy under midazolam sedation. The study also found that depression, endobronchial ultrasound-guided transbronchial needle aspiration (EBUS-TBNA), and high-dose midazolam administration were associated with disinhibition. However, it has been reported that EBUS-TBNA can be performed under conscious sedation with midazolam, with no complications and high patient satisfaction, using adequate surface anesthesia [84]. To avoid de-suppression during sedation, it is crucial to manage pain by providing sufficient local anesthesia.

### 3.3. Dexmedetomidine

Dexmedetomidine is an agonist of alpha 2 adrenergic receptors in the locus coeruleus and inhibits the activity of noradrenergic neurons in the central nervous system [85], which allows it to control fear and excitement and induce sedation with minimal respiratory depression. It is primarily used for intubated sedation in the ICU [86] and less commonly used for outpatient sedation because of its long elimination half-life [87]. However, combining dexmedetomidine with midazolam has also been studied to mitigate the disadvantages of dexmedetomidine [88]. Studies have also shown that dexmedetomidine can be used to provide sedation for implant surgery, resulting in lower pain levels and lower plasma levels of inflammatory cytokines than midazolam [89]. In a randomized controlled trial (RCT) comparing dexmedetomidine with midazolam, dexmedetomidine caused less anxiety at the same level of sedation [90]. Another RCT compared a combination of midazolam and dexmedetomidine to a combination of midazolam and propofol in terms of their effectiveness in preventing unexpected patient movements during dental surgery [91]. Therefore, dexmedetomidine is considered to provide stable and superior sedation during implant surgery. However, the long half-life of dexmedetomidine remains a concern for its use in sedation for dental treatment, as recovery time was not evaluated in these studies.

Owing to its “remarkably wide safety margins” [92], dexmedetomidine can be used in most dental patients. However, obesity can affect the effectiveness of sedatives, and although obesity itself does not affect the clearance of dexmedetomidine, obese patients tend to have higher plasma concentrations of the drug, suggesting that lean body mass should be used as a scaler for obese individuals [93]. Dexmedetomidine is metabolized in the liver through both glucuronidation and the cytochrome P450 system, and its clearance depends on hepatic blood flow, which can be impaired in patients with severe hepatic failure [94]. However, dexmedetomidine has also been suggested to exert a protective effect on the liver during hepatectomy [95]. Taken together, dexmedetomidine may safely provide stable sedation for implant surgery, although its long recovery time may be a concern for some patients and dentists.

### 3.4. Remimazolam

Remimazolam is a novel, ultra-short-acting intravenous benzodiazepine anesthetic [96]. According to a meta-analysis as a sedative for endoscopic procedures [97], remimazolam induces deeper sedation than midazolam but is slightly inferior to propofol. Additionally, it is safer to use than midazolam and propofol because of its minimal effects on respiratory and circulatory depression [97]. In a clinical study of patients with hepatic or renal impairment [98], the peak concentration after bolus intravenous injection of remimazolam was not affected by hepatic or renal impairment, the clearance of patients with severe hepatic impairment was reduced by 38.1%, and recovery was somewhat slower than in healthy subjects. In patients with renal impairment, plasma clearance was similar to that observed in healthy subjects. Remimazolam is metabolized by carboxylesterases 1A (CES-1A) in the hepatic metabolism, unlike other benzodiazepines, which are metabolized by the cytochrome p450 enzyme. Although CES-1A is known to be inhibited by alcohol, alcohol has reported to have no effect on remimazolam metabolism [99]. 

Furthermore, the effect of remimazolam can be reversed by flumazenil [100], a specific benzodiazepine receptor antagonist. A randomized controlled trial comparing the use of remimazolam with midazolam for sedation during oral surgery found that remimazolam resulted in a higher success rate and earlier recovery [101]. Despite limited published research on the use of remimazolam for sedation during implant surgery, it is expected to be a suitable sedative in clinical dental settings [102] (Table 2).

## 4. Practical Management

Patient satisfaction may be an ostensibly optimal indicator; however, it may simply reflect the depth of sedation attained. Conversely, surgeon satisfaction alone may not adequately reflect patient satisfaction. Both types of satisfaction are contingent on effective pain management. 

There is the concern of increased cardiovascular effects due to increased doses of adrenaline. Initial infiltration anesthesia is often insufficient for prolonged surgeries. Furthermore, for postoperative pain management, it is desirable for local anesthetics to remain effective for some time after surgery. If the patient does not have severe cardiovascular disease, infiltration anesthesia with a local anesthetic and adrenaline for hemostatic effects should be performed. For a prolonged effect, nerve blockade with ropivacaine or bupivacaine in conjunction with infiltration anesthesia can be used. Ultrasound-guided MNB, which is more reliable [37,38], is likely to become increasingly popular for pain control in dental implant surgery compared with conventional IANB. 

Sedation during implant surgery is intended to facilitate the procedure in a safe and efficient manner for both patients and surgeons rather than rendering the patient unconscious. For adult patients, adequate sedation is generally achieved when they achieve a score of 4 on the Observer’s Assessment of Alertness and Sedation (OAA/S) scale, indicating “lethargic responses to name spoken in normal tone” [90]. If a bispectral index (BIS) monitor is available, a BIS value of approximately 80 is generally considered to indicate adequate sedation for dental [90] and implant surgeries [54]. Combining BIS with target-controlled infusion (TCI) helps maintain a constant level of sedation during implant surgery [103]. A noteworthy suggestion is that “The sedation level was well-maintained within the range of conscious sedation in most cases” [2]. Thus, satisfactory sedation for patients and surgeons can be achieved with adequate local anesthetic effects and a stable level of conscious sedation.

Although adverse side effects are rare, extrapyramidal symptoms can be induced by midazolam injections intended for arousal sedation [104] and palliative care [105]. Acute dystonia has also been reported in a 6-year-old boy after midazolam injection for sedation [106]. Sexual hallucinations are a common issue associated with sedation [107]. Therefore, to prevent such occurrences and protect against potential patient complaints, it is recommended that extreme caution be exercised during any physical contact, and that one-on-one contact with patients be avoided.

Psychological stress caused by pain and discomfort during surgery can trigger physiological stress responses, including increases in the heart rate and blood pressure. Additionally, co-administration of adrenaline with local anesthetics can further activate adrenergic receptors. Moreover, co-administration of adrenaline with local anesthetics can further activate adrenergic receptors. Appropriate use of sedatives can effectively suppress these stress responses, promoting the stability of both heart rate and blood pressure, even with minimal doses of midazolam [41]. These findings suggest that sedation during implant surgery can improve safety by reducing stress responses, which may enhance the safety margin for vasoconstriction. Therefore, satisfactory sedation for both the patient and surgeon can also help ensure safety.

## 5. Conclusions

Comprehensive intraoperative pain management is necessary to ensure patient and surgeon satisfaction and safe implant surgery, which depends on the patient’s condition and the surgical procedure being performed. Maintaining a certain level of sedation under objective evaluation is necessary to reduce the discomfort associated with implant surgery. A more satisfactory management approach for both the patient and surgeon may result in safer implant surgery.

## Figures and Tables

**Table 1 jpm-13-00461-t001:** Circumstances in which the unique properties and methods of each local anesthetic can be employed. A: adrenaline, O: octapressin, IANB: inferior alveolar nerve block.

	Short Duration	Wide and Long Duration	Cardiovascular Diseases
Articaine+A [7,8,10] Lidocaine+A [9]	✔	✔	Dilution [18,19]
Propitocaine+O [11] Mepivacaine [13,14]	-	-	✔
Ropivacaine [20] Bupivacaine [21]	-	✔	✔
Intraosseous [34,35,36]	✔	-	-
Conventional IANB [31,32,33]	-	-	-
Echo-guided IANB using ropivacaine [37,38,39,40]		✔	✔
The main outcomes of the references are as follows: [7] Articaine is more effective than lidocaine in the first molar region during routine dental procedures. The side effects of both drugs appear to be similar (systematic review); [8] Articaine is more effective than lidocaine for local anesthesia for the treatment of pulpitis. Alticaine injections are less painful, more immediate, and have fewer adverse events when compared with lidocaine (umbrella review); [9] Four percent articaine 1:100,000 adrenaline was superior to two percent lidocaine, 1:100,000 adrenaline. Two percent lidocaine, 1:100,000 adrenaline was superior to three percent prilocaine, 0.03 IU ferypressin (systematic review); [10] The most effective local anesthetic for mandibular wisdom tooth extraction was 4% articaine, which was significantly more effective than 2% lidocaine, 0.5% bupivacaine, and 1% ropivacaine (meta-analysis); [11] Propitocaine with felypressin increased blood pressure, and lidocaine with adrenaline increased the heart rate; [13] Intraosseous injection of 2% lidocaine–adrenaline increased the heart rate but did not significantly increase the heart rate with intraosseous injection of 3% mepivacaine; [14] Compared with adrenaline-added lidocaine, 3% mepivacaine without vasoconstrictor had a significantly weaker local anesthetic effect, but it was better for patients with cardiac disease; [18] The effects of 2% lidocaine with 1:200,000 adrenaline were at the same level as that of 1:80,000 adrenaline, but 2% lidocaine with 1:80,000 adrenaline increased heart rate and blood pressure significantly; [19] The effects of anesthesia with 2% lidocaine and adrenaline concentrations of 1:50,000, 1:80,000, and 1:100,000 in the inferior alveolar nerve block were at the same level of success and failure; [20] Ropivacaine 0.75% resulted in a significantly longer duration of anesthesia and less intraoperative and postoperative analgesia than 2% lidocaine with adrenaline for implant surgery; [21] Bupivacaine with adrenaline is superior to lidocaine with adrenaline in relatively prolonged dental procedures, especially those requiring endodontic treatment or postoperative pain management (meta-analysis); [31] Both IANB and infiltration anesthesia are safe and effective for implant placement in the posterior mandible; however, IANB provides deeper analgesia than mandibular infiltration (RCT); [32] IANB may not be necessary for standard implant surgery in the posterior mandible, and infiltration with 4% articaine and 1:100,000 adrenaline may be sufficient (RCT); [33] IANB has been shown to fail in approximately 30% to 45% of cases, even when properly performed (review); [34] The combination of infiltration anesthesia, IANB, Vazirani–Akinosi nerve block, and IOI was more effective than IANB (meta-analysis); [35] Intraosseous injection with 2% lidocaine with adrenaline, 4% articaine with adrenaline, or buccal and lingual infiltration anesthesia with 4% articaine with adrenaline was significantly more effective for treatment of pulpitis in mandibular molars (meta-analysis); [36] For mandibular wisdom tooth extractions, intraosseous injection had significantly shorter anesthesia time than inferior alveolar nerve block (meta-analysis); [37] The mandibular nerve and its branches were stained with methylene blue in all cases of ultrasound-guided MNB via the lateral pterygoid approach in cadavers. No accidental injections into the facial nerve or maxillary artery were observed; [38] Ultrasound-guided alveolar nerve block (IANB) was effective for postoperative analgesia after osteomyelitis curettage for advanced drug-induced osteonecrosis of the jaw (MRONJ); [39] The efficacy of ultrasound-guided MNB compared with that of postoperative mandibular nerve block during mandibular fracture repair showed significant intraoperative and postoperative analgesia; [40] Ultrasound-guided MNB was performed in 217 patients who underwent maxillofacial surgery, with no reported complications.			

**Table 2 jpm-13-00461-t002:** Advantages of each sedative.

	Short-Acting	Antagonist	Less Respiratory Depression
Propofol [43,44]	✔✔	-	✔
Midazolam [1,70,71]	✔	✔	✔
Dexmedetomidine [85,86,87]	-	-	✔✔
Remimazolam [96,97,100]	✔✔	✔	✔
The main outcomes of the references are as follows: [1] Intraoperative heart rate and blood pressure decreased in the dexmedetomidine group during sedation for wisdom tooth extraction. Midazolam was associated with greater amnesia; [43] The pharmacokinetics of propofol were studied in 50 cases of general anesthesia. The mean systemic clearance rate of propofol was 2.09 ± 0.65 1/min (mean ± SD) and the elimination half-life was 116 ± 34 min; [45] The probability of cardiopulmonary complications was lower in sedation with propofol compared with conventional agents for colonoscopy (meta-analysis); [70] Flumazenil antagonizes the sedative effects of midazolam and has little effect on hemodynamic or respiratory kinetics (review); [71] Minimal oxygen saturation was significantly lower in the propofol group than in the midazolam group during sedation for thoracoscopy. Hypoxemia and hypotension were more common in the propofol group (RCT); [85] Dexmedetomidine is a selective α2-receptor agonist with sedative, analgesic, hypotensive, and bradycardic properties. Respiratory depression was minimal (review); [86] Dexmedetomidine significantly reduced the amount of concomitant alfentanil required for sedation in the ICU compared with propofol; [87] The pharmacokinetics of dexmedetomidine in patients managed in the postoperative ICU were similar to those previously observed in volunteers, with the exception of steady-state volume of distribution; [96] The pharmacokinetics of remimazolam showed that it had a rapid onset of effect and recovery, with some hemodynamic effects; [97] The sedative efficiency of remimazolam was significantly higher than that of midazolam but slightly lower than that of propofol. Inhibitory effects of remimazolam on respiration and circulation were weaker than midazolam and propofol (meta-analysis); [100] The recovery after remimazolam was much faster than that after midazolam administration. After flumazenil injection, the median awake time was reduced to 3.5 min, effectively restoring psychomotor and cardiovascular dysfunction.			

✔; applicable, ✔✔; strongly applicable.

## Data Availability

Not applicable.

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
