# Peer review of "Considerations for Satisfactory Sedation during Dental Implant Surgery"

_jpm, 2023, doi:10.3390/jpm13030461_

Round 1
Reviewer 1 Report (New Reviewer)
Comments to authors
Dear authors,
Thank you for submitting the manuscript entitled "Personalized management of pain and fear for dental implant surgery" for possible publication in the Journal of Personalized Medicine. This article can be interesting for the medical and dental community. However, I have some considerations related to the manuscript. You can find below my comments.
Title:
- I do not think the title of the paper really reflects its content. To be honest, when I receive this manuscript for review, I expected a broad range of interventions and approaches to control pain and fear from patient, which may involve psychological procedures, hypnosis, environmental changes in the dental office, etc. Nevertheless, your article only discusses the use of sedation to assist local anesthesia. Therefore, I kindly ask you to adjust the title to reflect the real content of your paper. Your manuscript basically describes the use of local anesthetics and conscious sedation in implant surgery.
Abstract:
- In the abstract, the authors mentioned the potential efficacy of “echo-guided inferior alveolar nerve block”. This term retrieved zero results when I put on Pubmed. So, please, replace this term in the whole manuscript for another one more frequently used.
- Keywords: I do not see a space for a “dental implantation” as a keyword of this article. I would remove this term and replace a more appropriate one (my suggestion: Literature Review).
Introduction:
- “Since implant surgery is an invasive treatment, it brings pain and fear to patients.” I completely disagree with this sentence. The interventions that involve “implant surgery” largely vary, and most of them are very securely performed without any “pain and fear” from patients. Recent studies have shown with patient-reported outcome measures that implant surgeries can be an almost painless procedure (Khouly et al. 2021). Please reconsider your affirmations in this regard.
- Line 32, “…a drilling in mandibular bone may induce severe pain.” Please, include a reference.
- An English review is necessary. There are some mistakes along the manuscript (ie, line 39, “Most of the patients who undergone implant treatment under sedation was hoping to get treated under sedation…” should be rephrased).
- “Several anesthetics are currently used for sedation of implants.” Are you sedating implants or patients? Please, consider a proof-reading in the whole manuscript to correct those mistakes.
- There are many sentences without references. For example, line 46, “Systemic disease, such as cardiovascular diseases, limits a use of vasoconstrictor added to local anesthetics.” Where are the original studies that confirm this affirmation?
Literature review section:
- Please clarify to readers the importance of sedation assessment at the beginning of the section “2. Assessment of Sedation”. When I first read, I had the impression those assessment had a minor importance in the context of pain and fear control.
- Table 1 should have an evidence support. Please, include the references that you used to build up this table. The authors might even consider to create a new table mentioning the studies and their outcomes.
- “Echo-guided IANB by using 5-6 ml of 0.375% of ropivacaine enabled to manage general anesthesia for mandibular sequestrectomy with lower dose of opioid and decreased a need of analgesics for 3 days after the surgery [39]. Besides, echo-guided IANB is reported to be able to perform safely [40]. Therefore, the echo-guided 189 IANB with long-acting local anesthetics is likely to be useful during and after the implant 190 surgery if it takes longer time.” Again, this “echo-guided” should be replaced by a more commonly used term such as “ultrasound-guided”. Moreover, the two references used are to support your affirmations are a retrospective study (39) and a case report (40), which means that they do not represent strong evidence for the above-mentioned sentences. Please rephrase the sentences dampened the affirmations.
- The same as for section “2. Assessment of Sedation”, please include some information on why the section “3.3. Metabolism of local anesthetics” is important to readers in the context of fear and pain control.
- Good review on sedatives.
- Table 2 can also benefit from references and a larger space, maybe including the original studies and their main results.
- After reading the manuscript, I wondered if this paper aimed to evaluate fear control of patients. If yes, please, clarify to readers. Include more references for the fear control too.
- Please include the paradoxical reaction of midazolam in your manuscript. Maybe a paragraph on it should be interesting. In addition, include at least a paragraph but maybe a whole section about the drawbacks and problems of conscious sedation.
- Please, acknowledge the problems of this “echo-guided” IANB. For me, a dental procedure that involves an EXTRAORAL injection may increase patient fear.
Conclusion:
- I do not see a room for including this echo-guided IANB in the conclusion section since the references presented by the authors are mainly low grade of evidence papers (retrospective studies and case report). Therefore, I would remove the ultrasound anesthesia from the Conclusion section.
Author Response
In Response to Reviewer #1’s Comments:
We greatly appreciate your insightful comments.- Title:
- I do not think the title of the paper really reflects its content. To be honest, when I receive this manuscript for review, I expected a broad range of interventions and approaches to control pain and fear from patient, which may involve psychological procedures, hypnosis, environmental changes in the dental office, etc. Nevertheless, your article only discusses the use of sedation to assist local anesthesia. Therefore, I kindly ask you to adjust the title to reflect the real content of your paper. Your manuscript basically describes the use of local anesthetics and conscious sedation in implant surgery.
According to comment, we changed the title, “Personalized management of Pain and Fear for Dental Implant Surgery”, to “Considerations for Satisfactory Sedation for Dental Implant Surgery”.
- Abstract:
- In the abstract, the authors mentioned the potential efficacy of “echo-guided inferior alveolar nerve block”. This term retrieved zero results when I put on Pubmed. So, please, replace this term in the whole manuscript for another one more frequently used.
- Keywords: I do not see a space for a “dental implantation” as a keyword of this article. I would remove this term and replace a more appropriate one (my suggestion: Literature Review).
According to comment, we changed the term, “echo-guided inferior alveolar nerve block”, to “ultrasound-guided mandibular nerve block” throughout whole manuscript. Also, a key word “dental implantation” was changed to Literature Review.
- Introduction:
- “Since implant surgery is an invasive treatment, it brings pain and fear to patients.” I completely disagree with this sentence. The interventions that involve “implant surgery” largely vary, and most of them are very securely performed without any “pain and fear” from patients. Recent studies have shown with patient-reported outcome measures that implant surgeries can be an almost painless procedure (Khouly et al. 2021). Please reconsider your affirmations in this regard.
The reference cited (Khouly et al., 2021) is a meta-analysis of postoperative analgesia and not directly relevant to the pain experienced during implant treatment. The study is not likely to assert that implant treatment is a virtually painless procedure. To date, only a limited number of studies have scrutinized the magnitude of pain during implant surgery and the impact of local anesthetics on it. The majority of studies, including the one cited, have centered on postoperative analgesia. Nevertheless, it is undisputed that implant surgery is invasive and accompanied by pain, warranting the use of local anesthesia. On the other hand, as reviewer have rightly observed, the extent of fear and anxiety among patients can vary based on their subjective perception of the treatment. Thus, I have accordingly revised this portion as follows,
“Implant surgery involves a variety of techniques and treatment strategies, and extensive and lengthy procedures often involve sedation to reduce discomfort. Even with sedation, the local anesthetic must be sufficiently effective to ensure patient satisfaction.” (line 27-29)
- Line 32, “…a drilling in mandibular bone may induce severe pain.” Please, include a reference.
The sentence is inappropriate. It was removed.
- An English review is necessary. There are some mistakes along the manuscript
According to comments, The English language has been checked by a specialist and generally corrected.
- There are many sentences without references. For example, line 46, “Systemic disease, such as cardiovascular diseases, limits a use of vasoconstrictor added to local anesthetics.” Where are the original studies that confirm this affirmation?
In response to comment, following references were added.
- Becker, D.E.; Reed, K.L. Local Anesthetics: Review of Pharmacological Considerations. Anesth Prog 2012, 59, 90–102, doi:10.2344/0003-3006-59.2.90.
- Gandy, W. Severe Epinephrine-Propranolol Interaction. Ann Emerg Med 1989, 18, 98–99, doi:10.1016/S0196-0644(89)80324-5.
- MASK, A.G. Medical Management of the Patient with Cardiovascular Disease. Periodontol 2000 2000, 23, 136–141, doi:10.1034/j.1600-0757.2000.2230114.x.
- Literature review section:
- Please clarify to readers the importance of sedation assessment at the beginning of the section “2. Assessment of Sedation”. When I first read, I had the impression those assessment had a minor importance in the context of pain and fear control.
I agree with the comments. The section was removed and parts of them was removed to “1. Introduction” (line 30-38) and “4. Practical Management” .
- Table 1 should have an evidence support. Please, include the references that you used to build up this table. The authors might even consider to create a new table mentioning the studies and their outcomes.
According to comments, references were added and we made Box1 and 2, in which outcomes in each reference were described.
- “Echo-guided IANB by using 5-6 ml of 0.375% of ropivacaine enabled to manage general anesthesia for mandibular sequestrectomy with lower dose of opioid and decreased a need of analgesics for 3 days after the surgery [39]. Besides, echo-guided IANB is reported to be able to perform safely [40]. Therefore, the echo-guided 189 IANB with long-acting local anesthetics is likely to be useful during and after the implant 190 surgery if it takes longer time.” Again, this “echo-guided” should be replaced by a more commonly used term such as “ultrasound-guided”. Moreover, the two references used are to support your affirmations are a retrospective study (39) and a case report (40), which means that they do not represent strong evidence for the above-mentioned sentences. Please rephrase the sentences dampened the affirmations.
According to comment, “echo-guided” was changed to “ultrasound-guided”. Other references were added for ultrasound-guided mandibular nerve block.
- The same as for section “2. Assessment of Sedation”, please include some information on why the section “3.3. Metabolism of local anesthetics” is important to readers in the context of fear and pain control.
I agree with comments, and removed both sections. Parts of description of each section were moved to line 30-38, line 106-114, and line 333-337.
- Good review on sedatives.
Thank you.
- Table 2 can also benefit from references and a larger space, maybe including the original studies and their main results.
Same as Table 1, reference numbers were added, and we added Box 2 for outcomes in each reference.
- After reading the manuscript, I wondered if this paper aimed to evaluate fear control of patients. If yes, please, clarify to readers. Include more references for the fear control too.
As suggested from reviewer, the word “fear” is inappropriate. Therefore, we changed to “discomfort”.
- Please include the paradoxical reaction of midazolam in your manuscript. Maybe a paragraph on it should be interesting. In addition, include at least a paragraph but maybe a whole section about the drawbacks and problems of conscious sedation.
According to comment, a paragraph was added (line 258-265).
- Please, acknowledge the problems of this “echo-guided” IANB. For me, a dental procedure that involves an EXTRAORAL injection may increase patient fear.
I agree with comment, and added sentences, “However, the procedure of inserting a needle from the inferior aspect of the zygomatic arch to its depths can be intimidating for patients. Therefore, it is crucial to comprehend their emotional state and provide information about various pain management techniques, allowing patients to make an informed choice.” (line 142-146)
- Conclusion:
- I do not see a room for including this echo-guided IANB in the conclusion section since the references presented by the authors are mainly low grade of evidence papers (retrospective studies and case report). Therefore, I would remove the ultrasound anesthesia from the Conclusion section.
According to comment, “ultrasound-guided mandibular nerve block” was removed from the Conclusion.
Reviewer 2 Report (New Reviewer)
In my opinion, the topic is one interesting.
Unfortunately, the objective and the result o this article are not quit clear presented and, may be the most important think it is important to clarify the way to evaluate the patient satisfaction in the same time with surgeons satisfaction. It is mandatory to be more clear in conclusions so to propose a clear way to increase, ion the same time, patient an surgeon's satisfaction
Author Response
In response to this comment, we revised a large portion of the manuscript and endeavored to describe management practices clearly that would enhance patient satisfaction.
Round 2
Reviewer 1 Report (New Reviewer)
Comments to authors
Dear authors,
Thank you for submitting the revised version of the manuscript entitled "Considerations for Satisfactory Sedation for Dental Implant Surgery" for possible publication in the Journal of Personalized Medicine. The article improved substantially from the previous version and I congratulate the authors for performing relevant modifications in the manuscript. However, some minor despite important points should be addressed before publication. Please, find below my comments.
Title:
- The title looks much better now. However, I suggest some grammatical alteration to suppress the repetition of “for”. Consider rephrasing to “Considerations for Satisfactory Sedation on Dental Implant Surgery”.
Main manuscript:
- Many sentences of grammatical concern can be observed. For example, line 50, “Implant surgery involves a variety of techniques and treatment strategies, and extensive and lengthy procedures often involve sedation to reduce discomfort.” There is an oddly and recurrent use of the conjunction “and”. Please, proofread the whole manuscript adjusting grammatically incorrect sentences accordingly.
- For the next round of revisions, please provide a “clean” version of the manuscript.
- When I mentioned the paradoxical reaction of midazolam, I was referring to the agitation that midazolam and other sedatives can provoke in some patients. Although infrequent, this side effect of midazolam administration has important implications since it can result in paradoxical manifestations of profound delirium with extrapyramidal symptoms, including hallucinations during conscious sedation for dentistry. Please expand the discussion of those side effects of conscious sedative use.

This manuscript is a resubmission of an earlier submission. The following is a list of the peer review reports and author responses from that submission.
Round 1
Reviewer 1 Report
The purpose by the study of Takaya Ito et al. (“Personalized Management of Stress Reaction During Dental 2 Treatment”) was to highlight the mechanism of stress reaction, the formation of fear memory, and differences in the effects of sedatives and local anesthetics. With that, the authors were unable to achieve the goals of the study.
There are several shortcomings that significantly affected the quality of the study.
Part 2 of "Stress reaction and its mechanism" is a superficial outline of the physiological systems involved in the implementation of the stress response. There is no information on the change in action of the same systems against the background of the introduction of anesthesia, which could serve as a link to the second (dental) part of the study (DOI: 10.1055/s-2007-1000588; doi: 10.1016/j.bjae.2020.04. 006; DOI: 10.4103/1119-3077.134058 etc).
The authors have not adequately and unsystematically described the well-known types of sedation used in dentistry for several decades. There is no classification of sedation according to the type of sedative delivery: oral sedation, IV sedation, and inhalation sedation. In the oral sedation group, the authors describe only one drug from the benzodiazepine group, while not specifying other oral drug options that are used for sedation. The advantages and disadvantages of the known options are not indicated. There is no indication of the type of oral sedatives: in syrups, tablets, drops, etc. - their features, indications, and contraindications. The group of inhaled sedatives is not mentioned in principle by the authors. A similar incomplete and inconsistent description concerns IV sedation.
Author Response
In Response to Reviewer #1’s Comments: We greatly appreciate your insightful comments. 1. Part 2 of "Stress reaction and its mechanism" is a superficial outline of the physiological systems involved in the implementation of the stress response. There is no information on the change in action of the same systems against the background of the introduction of anesthesia, which could serve as a link to the second (dental) part of the study (DOI: 10.1055/s-2007-1000588; doi: 10.1016/j.bjae.2020.04. 006; DOI: 10.4103/1119-3077.134058 etc). According to this comment, we have added the “1) Usefulness of sedation” in “3. Individual differences in the effects of sedatives” “Propofol significantly reduces hemodynamic and metabolic changes caused by chest physical therapy in postoperative patients in the intensive care unit52. Comparison of changes in hemodynamic parameters and level of stress hormones, including blood glucose and serum adrenaline, between propofol with dexmedetomidine in patients undergoing cholecystectomy under general anesthesia showed better effects of propofol on decreasing blood glucose and serum adrenaline levels53. However, the direct effect of propofol on cortisol synthesis from cultured bovine adrenocortical cells is less specific than that of etomidate and thiopentone54. Midazolam has also been reported to control transient hypertension immediately before dental treatment55. The sublingual administration of midazolam is also effective in controlling the level of salivary cortisol before general anesthesia for the extraction of the third molars56. However, since a low dose of midazolam can lead to a slight increase or almost no change in blood pressure5758, the effect of midazolam on blood pressure is not considered a direct action on the cardiovascular system. Both propofol and midazolam exert anesthetic effects by activating GABA receptors. Therefore, the effects of both anesthetics against stress reactions are not likely direct actions on the adrenal gland; rather, these anesthetics suppress the input signal to the PVN, leading to controlled activation of the HPA axis.” 2. The authors have not adequately and unsystematically described the well-known types of sedation used in dentistry for several decades. There is no classification of sedation according to the type of sedative delivery: oral sedation, IV sedation, and inhalation sedation. In the oral sedation group, the authors describe only one drug from the benzodiazepine group, while not specifying other oral drug options that are used for sedation. The advantages and disadvantages of the known options are not indicated. There is no indication of the type of oral sedatives: in syrups, tablets, drops, etc. - their features, indications, and contraindications. The group of inhaled sedatives is not mentioned in principle by the authors. A similar incomplete and inconsistent description concerns IV sedation. According to this comment, we have added the headings “1) Usefulness of sedation”, “2) Intravenous anesthetics”, “3) Inhalational anesthetics”, and “4) Oral sedatives”, and have also added explanations for 3) and 4) to “3. Individual differences in the effects of sedatives”.
Reviewer 2 Report
This review describes the mechanism of stress reaction, formation of fear memory, and differences in the effects of sedatives and local anesthetics as the patients undergo dental treatment. Moreover, this review described the complex roles played by the central nervous system, the advantages, and disadvantages of propofol and midazolam, commonly used sedatives, as well as the indications and limitations of commonly used local anesthetics, were elaborated in detail, which pointed out the direction for clinical drug use and provided certain theoretical guidance for preventing and reducing patients' tension and psychological pressure during oral medical treatment.
However, the article didn’t provide enough content about the connection between dental treatment and stress reaction. For example, on page 4, the manuscript described that “the activation of the SNS is physiologically normal in response to internal and/or external changes, it can result in severe outcomes in vulnerable persons” in detail, but the description of the “dental treatment stimulating the SNS may deteriorate cardiovascular diseases in patients with vulnerable cardiovascular conditions” is too simple to clearly show the effects caused by the dental treatment to the cardiovascular diseases.
The authors described the mechanism of stress response and the difference between the effects of sedatives and local anesthetics. The title of this review is “personalized management of stress reaction during dental treatment”, while the content about “personalized management” is lack of in the article, the author didn’t describe the specific preventive measures for the patients undergoing dental treatment. Besides, more recommendations and future expectations on the application of drugs are needed in the conclusion.
Author Response
In Response to Reviewer #2’s Comments: We appreciate your constructive comments. 1. However, the article didn’t provide enough content about the connection between dental treatment and stress reaction. For example, on page 4, the manuscript described that “the activation of the SNS is physiologically normal in response to internal and/or external changes, it can result in severe outcomes in vulnerable persons” in detail, but the description of the “dental treatment stimulating the SNS may deteriorate cardiovascular diseases in patients with vulnerable cardiovascular conditions” is too simple to clearly show the effects caused by the dental treatment to the cardiovascular diseases. According to these comments, we have revised the sentence “such as person with heart disease, psychological disease, or CPVT” as follows, “Therefore, even in dental treatment, SNS stimulation may deteriorate cardiovascular diseases in patients with vulnerable cardiovascular conditions, such as heart disease, psychological disease, or CPVT.” 2. The authors described the mechanism of stress response and the difference between the effects of sedatives and local anesthetics. The title of this review is “personalized management of stress reaction during dental treatment”, while the content about “personalized management” is lack of in the article, the author didn’t describe the specific preventive measures for the patients undergoing dental treatment. Besides, more recommendations and future expectations on the application of drugs are needed in the conclusion. Thank you for your suggestion. To emphasize “personalized management,” various parts of the Introduction have been revised as follows: “the effects of sedatives differ among patients depending on their medical condition.” “However, since each formulation has a slightly different effect and the risk of complications increases as the dosage increases, they can be used more effectively by considering their usage according to the treatment and the patient's condition.” “Thus, stress reactions to dental treatment vary among individuals and a variety of factors influence the effectiveness of sedation. Local anesthetics are generally safe; however, as the dose increases, individualized management is necessary. Thus, personalized management is necessary because the stress response differs depending on the patient's background and the nature of the treatment.” In response to your suggestion, we have rewritten the entire Conclusion.
Round 2
Reviewer 1 Report
The authors of the manuscript did some work to correct for it.
However, the material presented is poorly structured and incomplete with a fragmented description of some of the sedation methods used in a dental appointment. The description of existing methods of sedation remains incomplete, despite the fact that this was pointed out in the first round of the review.
The title of the manuscript on personalized management of stress responses at the dental appointment does not correspond to the descriptive part of the manuscript.
The scientific novelty of the research is absent. There is no systematism in the presentation of the material. The text in its current presentation is extremely difficult to read and understand.
Author Response
Thank you for your valuable comments. Your comments helped us to improve our manuscript greatly. I have rewritten it extensively with the intention of correcting the points the reviewer has raised. I hope that this revised manuscript would be satisfactorily meeting the comments.
In the chapter "2. Stress Reactions and Their Mechanisms", the text was substantially rewritten. Section headings were removed and unnecessary sentences were removed for academic papers. In the chapter "3. Individual differences in sedative effects", we added age-specific explanations to the sections on propofol and midazolam. In addition, sections on dexmedetomidine, remimazolam, inhalation anesthetics, and oral sedatives were added. Furthermore, a section on "Relationship between Anesthetic Action and Stress Response" was added in the chapter. Finally, we added a chapter on "5. Optimizing the management" and its schema to present an overview of this paper.
Reviewer 2 Report
The author added a little content of “personalized management” into the text to describes the mechanism of stress reaction, formation of fear memory, and differences in the effects of sedatives and local anesthetics. The conclusion was also revised. The content about “personalized management” is still lack of in the revised article.
Author Response
Thank you for your valuable comments. They helped us to improve our manuscript greatly. In response to comments, unnecessary sentences were deleted and an explanation of "personalized management" was added. I hope that this revised manuscript would be satisfactorily meeting the comments.
In the chapter "2. Stress Reactions and Their Mechanisms," the text was substantially rewritten. Section headings were removed and unnecessary sentences were removed for academic papers. In the chapter "3. Individual differences in sedative effects," we added age-specific explanations to the sections on propofol and midazolam. In addition, sections on dexmedetomidine, remimazolam, inhalation anesthetics, and oral sedatives were added. Furthermore, a section on "Relationship between Anesthetic Action and Stress Response" was added in the chapter. Finally, we added a chapter on "5. Optimizing the management" and its schema to present an overview of this paper.